# miRNAs and Hematological Markers in Non-Alcoholic Fatty Liver Disease—A New Diagnostic Path?

**DOI:** 10.3390/biomedicines13010230

**Published:** 2025-01-18

**Authors:** Agata Michalak, Małgorzata Guz, Joanna Kozicka, Marek Cybulski, Witold Jeleniewicz, Ilona Telejko, Karolina Szczygieł, Ewa Tywanek, Halina Cichoż-Lach

**Affiliations:** 1Department of Gastroenterology, Medical University of Lublin, Jaczewskiego 8, 20-954 Lublin, Poland; joanna.kozicka@umlub.pl (J.K.); halina.lach@umlub.pl (H.C.-L.); 2Department of Biochemistry and Molecular Biology, Medical University of Lublin, Chodźki 1, 20-093 Lublin, Poland; malgorzata.guz@umlub.pl (M.G.); marek.cybulski@umlub.pl (M.C.); witold.jeleniewicz@umlub.pl (W.J.); ilona.telejko@umlub.pl (I.T.); 3Clinical Dietetics Unit, Department of Bioanalytics, Medical University of Lublin, Chodźki 7, 20-093 Lublin, Poland; karolina.szczygiel@umlub.pl; 4Department of Internal Medicine and Internal Medicine in Nursing, Medical University of Lublin, Chodźki 7, 20-093 Lublin, Poland; ewa.tywanek@gmail.com; 5Department of Endocrinology with Nuclear Medicine Department, Center of Oncology of the Lublin Region St. Jana z Dukli, Jaczewskiego 7, 20-090 Lublin, Poland

**Keywords:** liver steatosis, NAFLD, liver disorders, miRNAs, diagnostic markers, hematological indices

## Abstract

**Background:** Asymptomatic liver steatosis constitutes an emerging issue worldwide. Therefore, we decided to explore relationships between selected types of microRNAs (miRNAs), serological markers of liver fibrosis and hematological parameters in the course of non-alcoholic fatty liver disease (NAFLD). **Methods:** Two hundred and seven persons were included in the survey: 97 with NAFLD and 110 healthy controls. Serological concentrations of miR-126-3p, miR-197-3p, and miR-1-3p were measured in all participants. Direct indices of liver fibrosis [procollagen I carboxyterminal propeptide (PICP), procollagen III aminoterminal propeptide (PIIINP), platelet-derived growth factor AB (PDGF-AB), transforming growth factor-α (TGF-α) and laminin] together with indirect markers (AAR, APRI, FIB-4 and GPR) were also evaluated. The assessment of hematological parameters concerned: mean platelet volume (MPV), platelet distribution width (PDW), plateletcrit (PCT), red blood cell distribution width (RDW), MPV to platelet (PLT) ratio (MPR), RDW to PLT ratio (RPR), neutrophil to lymphocyte (LYM) ratio (NLR), PLT to LYM ratio (PLR) and RDW to LYM ratio (RLR). Additionally, the NAFLD fibrosis score and BARD score were applied. **Results:** The concentration of miR-126-3p and miR-1-3p was higher, and miR-197-3p was lower in the NAFLD group (*p* < 0.0001). miR-197-3p correlated notably with hematological indices: negatively with PDW (*p* < 0.05) and positively with PLR (*p* < 0.05). **Conclusions:** Significant correlations between miRNA molecules and hematological markers in the course of NAFLD indicate inflammation as a potential background and create new possibilities for a diagnostic approach.

## 1. Introduction

Non-alcoholic fatty liver disease (NAFLD) has become a global threat nowadays, with a prevalence of 29.8% worldwide. Future perspectives are even more alarming; by 2030, NAFLD might constitute the main cause of liver failure requiring its transplantation [1]. It is known as a pathology presenting with excessive fat deposition in hepatocytes that cannot be attributed to alcohol or other potential causative factors. Due to the significant heterogeneity of NAFLD, two other conditions concerning liver steatosis were formed lately: metabolic dysfunction-associated fatty liver disease (MAFLD) and metabolic dysfunction-associated steatotic liver disease (MASLD), highlighting a complex background of the steatosis accompanying obesity, insulin resistance and underlying cardiometabolic disorders [2]. Currently, NAFLD, MAFLD and MASLD are perceived as three independent pathologies. MAFLD is associated with metabolic risk abnormalities, and MASLD—with cardiometabolic criteria [3]. Excluding clinical differences, NAFLD, MAFLD and MASLD share excessive deposition of fat within hepatocytes (>5% of hepatocytes with macrovesicular steatosis and visible intracellular triglycerides or steatosis concerning at least 5% of the liver volume/weight). Due to the lack of liver steatosis’ certain symptoms and the appearance of its clinical manifestation in the stage of already developed advanced steatohepatitis and liver cirrhosis, NAFLD patients requiring medical support become relatively often overlooked. Another danger ahead constitutes the risk of developing hepatocellular carcinoma (HCC)—even in the pre-cirrhotic stage. To avoid these unpleasant and mostly irreversible consequences, it is worth searching for new non-invasive indices of steatosis in order to screen the general population. Even though laboratory tests and imaging tools are used in clinical settings to diagnose and monitor NAFLD or to predict the risk of liver fibrosis in its course, these solutions are imperfect, and liver biopsy still remains the gold standard for a final assessment. Novel non-invasive biomarkers, preferably obtained from the blood, would fill the diagnostic gap, improving the overall diagnosis of NAFLD. Such candidates could be microRNAs (miRNAs), regulatory particles mediating neoplastic process, inflammation or autoimmunity. Simultaneously, they appeared to be involved in various pathologies related to liver disorders. Metabolic phenomena, thus enzyme cycles involving glucose and lipids, constitute biochemically a background inseparably related to miRNAs and their regulation. Therefore, these molecules have been explored in the course of NAFLD, alcohol-related liver disease and diabetes [4]. Within the liver, miRNAs were found to affect the function of various liver cell types, including hepatocytes, stellate cells and Kupffer cells [5]. Being one of the most frequent types of miRNAs in the liver, miR-122 was explored as a potential biomarker of NAFLD and its inhibition was related to the protection of hepatocytes from the disease [6]. In contrast, in mice models with miR-223-knockout, the development of NAFLD was observed [7]. On the other hand, hematological indices are explored in the numerous fields of medicine as prognostic markers of chronic diseases or acute pathologies [8]. Some of them were even proposed as potential markers of liver steatosis. According to the available literature, miR-1-3p, miR-126-3p and miR-197-3p seem to be uninvestigated in NAFLD patients. Moreover, it is barely impossible to find in the literature relationships between already well-known markers of liver fibrosis (direct and indirect parameters), hematological indices and miRNA molecules.

Therefore, we decided to explore in our study the diagnostic accuracy of miR-126-3p, miR-197-3p and miR-1-3p in the course of NAFLD and to identify existing relationships between them and other markers.

## 2. Materials and Methods

We recruited 207 persons to the survey: 97 patients with NAFLD and 110 healthy people (no chronic disorders or known liver failure) in the control group. The research group was formed by patients of the Gastroenterology Department with the Endoscopy Unit of the Medical University of Lublin and the Gastroenterology Outpatient Clinic. Participants were included in the study from 2017 to 2020. The clinical features of the study participants are shown in Table 1, and their biochemical characteristics, together with the scales’ applied scores, are presented in Table 2. NAFLD was diagnosed based on fatty liver parenchyma visualized on the ultrasound examination and a negative history of alcohol use. Patients reported occasional alcohol consumption of no more than 10 g of pure alcohol per day or declared complete abstinence. A liver biopsy was performed on 43 people: 30 patients had fatty liver disease, and 13 patients had steatohepatitis. Forty-eight patients underwent FibroScan examination; liver fibrosis was excluded (METAVIR F0). Panendoscopy was performed in all subjects; no signs of portal hypertension were found. Twenty-seven people were diagnosed with diabetes mellitus type 2 (DM2). Patients who have type 1 diabetes were excluded from the study. Abnormal fasting glucose was not detected in any person other than diabetics. Fifty-three patients with NAFLD had arterial hypertension (HT), and 90 people were diagnosed with metabolic syndrome (MS), the diagnosis of which is based on meeting at least three of the following five criteria: abdominal obesity (waist circumference: in men ≥ 94 cm, in women ≥ 80 cm), TG ≥ 150 mg/dL or treatment of dyslipidemia, HDL cholesterol < 40 mg/dL in men, <50 mg/dL in women or treatment of dyslipidemia, blood pressure ≥ 130/85 mm Hg or treated HT, fasting blood glucose ≥ 100 mg/dL or DM2 treatment. To eliminate other potential etiological factors of liver diseases, serological indicators for viral damage (hepatitis A/B/C virus, Epstein–Barr virus, cytomegalovirus) and autoimmune pathologies (antinuclear antibodies, antimitochondrial antibodies, anti-smooth muscle antibodies, antibodies against liver–kidney microsomes) were tested in all of the study group, obtaining negative results. The concentration of alkaline phosphatase in the studied patients was within the reference range. The presence of clinically significant acute or chronic inflammation was also excluded. None of the people enrolled in the study took steroids. Each person gave written consent to participate in the survey. The investigation was carried out in accordance with the protocol approved by the Bioethics Committee of the Medical University of Lublin (decision no. KE-0254/86/2016).

Serological concentration of miR-126-3p, miR-197-3p and miR-1-3p was determined in all studied patients [miRNA enzyme immunoassay (miREIA); BioVendor, Brno, Czech Republic]. Direct parameters of liver fibrosis [procollagen I carboxyterminal propeptide (PICP), procollagen III aminoterminal propeptide (PIIINP), platelet-derived growth factor AB (PDGF-AB), transforming growth factor-α (TGF-α) and laminin] together with indirect indices (AAR, APRI, FIB-4 and GPR) were also investigated. To assess the expression of direct indices, ELISA kits were used [PICP, PIIINP—Wuhan EIAab Science (Wuhan, China), PDGF-AB, TGF-α—R&D Systems Quantikine ELISA Kit (Minneapolis, MN, USA), laminin—EIA Kit without Sulphuric Acid (Kusatsu, Shiga, Japan)]. The evaluation of hematological parameters included mean platelet volume (MPV), platelet distribution width (PDW), plateletcrit (PCT), red blood cell distribution width (RDW), MPV-to-platelet (PLT) ratio (MPR), RDW-to-PLT ratio (RPR), neutrophil-to-lymphocyte (LYM) ratio (NLR), PLT-to-LYM ratio (PLR) and RDW-to-LYM ratio (RLR).

### Statistical Analysis

Statistical analysis was performed using Statistica version 13.3 (TIBCO Software Inc.; Santa Clara, CA, USA) software for Windows. Deviation from normality was calculated with the Kolmogorov–Smirnov test. The Mann–Whitney U test was performed for between-group comparisons due to non-normal distribution. Spearman correlation analyses were carried out to evaluate the correlations. All probability values were two-tailed, and a value of *p* less than 0.05 was described as statistically significant. The positive predictive value (PPV) and negative predictive value (NPV) of the examined markers were also assessed. Receiver operating characteristic (ROC) curves and area under the curve (AUC) values were obtained to determine the sensitivity, specificity and proposed cut-offs of investigated miRNAs in MASLD groups. ROC analysis was performed using Medical Bundle for Statistica software. Optimal cut-off values for miRNA levels were established using the tangent method.

## 3. Results

The results of the examined miRNAs (miR-126-3p, miR-197-3p and miR-1-3p) are shown below in Table 3. Analysis revealed a significantly higher serological expression of miR-126-3p and miR-1-3p in NAFLD patients compared to the control group (*p* < 0.0001); the concentration of miR-197-3p was notably lower (*p* < 0.0001).

The results of assessed hematological parameters are presented in Table 4. The median values of all evaluated parameters (except MPR and PCT) showed statistically significant differences compared to controls; the medians of PLT, MPV and LYM were notably lower (*p* < 0.001, *p* < 0.0001, *p* < 0.0001, respectively), while the medians of PDW, RDW, RPR, NEU, NLR, PLR and RLR had significantly higher values (*p* < 0.01, *p* < 0.05, *p* < 0.0001, *p* < 0.05, *p* < 0.0001, *p* < 0.05, *p* < 0.0001, respectively).

The results of determined indirect parameters of liver fibrosis in examined persons are presented in Table 5. Patients with NAFLD were characterized with notably higher median values of APRI, FIB-4 and GPR in comparison to the control group (*p* < 0.0001); the result of AAR was not statistically different.

Table 6 shows the serological expression of direct markers of liver fibrosis in the studied groups. Among them, medians of TGF-α and laminin were notably lower compared to controls (*p* < 0.0001) and the median of PDGF-AB—higher (*p* < 0.05).

Statistically significant mutual relationships between miRNAs and other assessed parameters are illustrated in Table 7. In the group of NAFLD patients, statistically notable relationships were found between miR-197-3p and hematological markers. miR-197-3p molecule correlated negatively with PDW and positively with PLR (*p* < 0.05).

The AUC values of all assessed markers among study participants are presented in Table 8 below. Among assessed miRNAs, miR-126-3p was characterized with the highest (but still modest) diagnostic accuracy, with an AUC value of 0.716 (*p* < 0.0001); the AUC value of miR-197-3p was 0.691 (*p* < 0.0001). Because of the notably low serous concentration of miR-1-3p in the study group and controls, it was not possible to verify its AUC in studied patients.

Table 9 visualizes calculated diagnostic cut-off points of explored biomarkers, their sensitivity and specificity, together with PPV and NPV. Figure 1 and Figure 2 show ROC curves for miR-126-3p and miR-197p, respectively, in examined NAFLD patients.

## 4. Discussion

Many attempts have been made so far to place miRNAs in clinical settings. Nevertheless, they still seem to play the greatest role in clinical trials, where their participation in oncogenesis, the development of autoimmunity or oxidative stress, was already confirmed. Simultaneously, surveys exploring hepatic pathologies related to the disturbances of miRNAs indicate acute liver failure, liver fibrosis and the development of HCC as essential examples. Nevertheless, there are still areas in hepatology where certain miRNA molecules have not been tested so far [9]. Such a gap can be observed in reference to NAFLD and three types of miRNAs investigated in the current study: miR-126-3p, miR-197-3p and miR 1-3p. A technique applied here to verify their serous concentration was miREIA. It is based on the hybridization of miRNA isolated from a patient sample to a complementary biotinylated DNA oligonucleotide probe. Obtained DNA/RNA hybrids bind to immobilized on the plate specific monoclonal antibodies, and further steps follow standard enzyme-linked immunosorbent assay (ELISA) protocols. miREIA allows for the evaluation of direct types of miRNA without the need for their prior mapping/sequencing using polymerase chain reaction (PCR). We used a similar method in the evaluation of the same miRNAs among patients with alcohol-related liver cirrhosis (ALC), published recently [10]. NAFLD-associated fibrosis constitutes a global problem nowadays; therefore, from a clinical point of view, the introduction of non-invasive (preferably laboratory) screening tools in the diagnosis of it would be a milestone in medicine. It is worth exploring potential combinations of different markers according to their various relationships with the background of NAFLD. Such compositions could also present a greater overall diagnostic accuracy. Nevertheless, the number of such studies in the current literature can be described as very low. Some investigations were even looking for significant dependences between miRNAs and other serological parameters as biomarkers of NAFLD; miR-122 and pro-neurotensin were examined, revealing their notably higher concentrations in NAFLD patients [11]. To broaden such a diagnostic path, the major goal of our survey was to verify the accuracy of the selected miRNAs in NAFLD patients together with their mutual correlations with other laboratory markers.

In one of the latest studies, miR-126-3p was described as a potential marker of subclinical atherosclerosis in obese patients, myocardial infarction and general cardiovascular risk [12,13,14]. The upregulation of miR-126-3p was additionally demonstrated in patients with proliferative diabetic retinopathy [15]. Its involvement in oncogenesis was also highlighted, e.g., in cholangiocarcinoma, lung and prostate cancer [16,17,18]. On the other hand, it has recently emerged that miR-126-3p is capable of stimulating natural killer cells in liver tumors, which could potentially be used in oncological treatment [19]. Simultaneously, miR-126-3p was found to interact with hepatic stellate cells in liver cancer [20]. Its downregulation was also found in the course of axial spondyloarthritis [21]. Regarding hepatology, current reports concern the participation of miR-126-3p, mainly in liver cancer. According to available literature, there are no available data on certain relationships between NAFLD and miR-126-3p. Our results showed its relatively high diagnostic accuracy in the course of liver steatosis. A metabolic background can be perceived as a link between a higher concentration of miR-126-3p in NAFLD patients from our survey and in the population of people with cardiovascular risk [22]. Previously, miR-126-3p was described as a particle involved in oxidative stress [23]. Therefore, relationships demonstrated in our study might also be related to an ongoing inflammation in NAFLD. The current experience concerning miR-197-3p involves its role in cancer as a biomarker of the response to treatment [24]. So far, there are just several available reports exploring miR-197-3p in the course of liver pathologies. Interestingly, many of them present the involvement of this molecule in liver steatosis by affecting the function of stellate cells or the expression of HCG18 in NAFLD patients [25,26]. Others showed the participation of miR-197-3p in the process of liver fibrosis in chronic HCV infection and liver cancer [27,28]. Our survey presented a notably lower concentration of miR-197-3p in NAFLD patients compared to controls, which seems to be confusing according to already gathered data. On the other hand, this molecule is correlated with hematological indices, such as PDW and PLR. The role of miR-1-3p was mainly explored in oncology (e.g., lung, breast and prostate cancer) [29,30,31]. In the field of hepatology, miR-1-3p was presented to participate in the development of HCC [32,33]. In our study, its expression was too low among all study participants to assess the diagnostic accuracy (AUC) of this particle in NAFLD; nevertheless, the concentration of miR-1-3p in the course of NAFLD in comparison to controls was notably higher.

Another important part of our investigation was to evaluate the existing relationships between miRNAs, direct and indirect markers of liver fibrosis and hematological indices. Our NAFLD patients presented notable correlations between miR-197-3p and both PDW and PLR. Hematological indices are still present in various clinical studies according to numerous pathologies: cardiovascular complications, oncogenesis, autoimmunity and liver disorders [34,35,36,37,38]. These explorations are usually related to the inflammatory background of these pathologies. Therefore, liver cirrhosis or liver steatosis are also involved here. Higher values of PLR were observed in HCV patients with poorer outcomes [39]. Of note, NLR and PLR were even investigated as potential predictors of cardiovascular mortality in the course of MASLD [40]. Moreover, PLR was identified as the predictor of developed HCV infection in one of the recent studies [41]. NAFLD constitutes a subsequent hepatological area where PLR was also examined [42,43,44]. Thus, a correlation of this marker with miR-197-3p confirms an existing involvement of examined parameters in NAFLD. PDW constitutes another indicator, which was explored in clinical settings resembling, to some extent, studies involving PLR (myocardial infarction, ischemic stroke, autoimmune liver diseases, Hashimoto’s thyroiditis and cancer) [45,46,47,48,49,50]. It was also investigated among NAFLD patients, achieving higher levels in the course of steatohepatitis and liver fibrosis [51,52,53]. Furthermore, the combination of increased levels of both MPV and PDW was another feature demonstrated to be characteristic of NAFLD [54]. On the other hand, in patients with liver cirrhosis, both PDW and PLR served the role of predictors due to complications of the disease (hepatorenal syndrome, varices, bacterial peritonitis) [55,56]. Correlations observed in our study appear to confirm the involvement of miR-197-3p, PLR and PDW in NAFLD, and according to the literature, they seem to be novel findings. Simultaneously, except for certain indices of morphotic blood elements, platelets are well known to mediate inflammation via the release of various cytokines and this pathological pathway participates in the development of cardiovascular disorders (e.g., atherosclerosis, vascular remodeling) [57]. Deviations among platelets and their specific parameters in NAFLD should be investigated as potential biomarkers promoting the risk of MAFLD and MASLD. Platelet granules contain profibrotic particles and growth factors, which mediate the function of stellate cells. It can be suspected that the significantly increased serological expression of PDGF-AB in our NAFLD group compared to controls is related to the aforementioned properties of platelets, as they are one of the biological sources of this factor. Of note, some attempts were even made to explore the role of antiplatelet drugs in MASLD patients in order to prevent the development of HCC [58]. Another observation on mice revealed that platelet-derived thrombospondin might promote diet-induced NASH and liver fibrosis [59]. In our study group, the diagnostic accuracy of PLT was modest, with an AUC of 0.646. On the other hand, mean platelet volume (MPV) was characterized by the highest diagnostic accuracy among assessed hematological markers (0.842), and its value was significantly higher in the course of NAFLD than in the control group. Simultaneously, we did not observe any notable correlations between MPV and other parameters investigated in the survey. However, it is worth emphasizing that already obtained general data on MPV highlight its role in the increased probability of cardiovascular complications among NAFLD patients (e.g., atherosclerosis, thrombosis, stroke and myocardial infarction) [60,61]. A higher level of MPV was also presented as the probable marker with an independent predictive value for the development of NAFLD [62].

The future appears to belong to combinations of various markers in order to stratify the risk of the development of liver steatosis. However, we are still in front of selecting the most appropriate parameters. On the other hand, the AUC value of the combination of miR-126-5p and leptin in the diagnosis of steatosis was 0.950 in one of the recent studies [63]. Other authors explored the combination of serum miR-145-3p, 122-5p, 143-3p, 500a-5p, and 182-5p as a potential tool to predict the presence of liver steatosis [64]. In our study, we did not aim to create a model indicating the probability of liver steatosis. Our major idea was to evaluate the diagnostic accuracy of the selected types of miRNAs in the course of NAFLD, which turned out to be modest. Subsequently, we were curious about existing potential relationships between miRNAs and other more common markers in hepatology. Of note is that our recent research performed among patients suffering from ALC, evaluating the same types of miRNAs, serological indices, and hematological parameters, revealed the presence of definitely different correlations. In the course of ALC, significant relationships were noticed mainly between miRNAs and direct markers of liver fibrosis [10]. On the other hand, in the current survey, miRNA particles correlated only with hematological parameters. It can be suspected that observed discrepancies are inseparably related to certain pathology of the liver—steatosis or fibrosis. The investigated relationships are probably related to the inflammatory background of steatosis, too. A common umbrella of inflammation tends to be described as a pathological path shared by vascular disorders, obesity and MASLD [65]. Furthermore, sepsis was also explored as a condition ongoing with a significantly changed expression of miRNAs. miRNAs were even suggested to be potential markers of sepsis because of their involvement in the regulation of gene expression and their relatively stable character in biological fluids, including blood [66,67,68]. Their overexpression or downregulation was described in patients with sepsis-induced cardiac dysfunction, kidney injury or encephalopathy. Certain types of miRNAs may behave as both triggering and protective factors in the process of sepsis [69,70]. Of note, particles belonging to the families of miRNAs examined in our study were already tested according to their involvement in sepsis, as well. The overexpression of miR-126 was found to prevent brain injury in patients with a damaged blood–brain barrier by the inhibition of nuclear factor-κB signaling [71]. Another group of septic patients presented the downregulation of miR-126-3p, which even correlated notably with the severity of the disease [72]. miR-126-5p, belonging to the miR-126 family, was investigated in the course of sepsis-induced acute lung injury, presenting a lower expression [73]. miR-197 was also explored in sepsis settings. It turned out that its downregulation might exert a protective role concerning cardiomyocyte injury [74]. On the other hand, miR-1-3p was shown to participate in intestinal epithelial inflammation in the course of sepsis [75]. Due to the above-mentioned inflammatory context related to the changes in the expression of miRNAs, patients presenting features of clinically relevant inflammation were excluded from our study. However, NAFLD per se constitutes a pathology ongoing with inflammatory disturbances. Regarding obesity, it was revealed in one of the mice models several years ago that the lowered expression of miR-26a was significantly related to a marked chronic inflammation in chondrocytes [76]. Identifying certain genetic disturbances leading to obesity and MASLD could constitute the breakthrough in the management of metabolic disorders nowadays. Some attempts in this field showed a possible impact of miRNAs on the insulin-like growth factor-1 signaling pathway [77]. Moreover, targeting certain miRNAs was proven to exert a similar effect on MAFLD patients as lifestyle modification based on physical exercises [78]. A notable involvement of miRNAs and hematological indices in the course of NAFLD among patients investigated in the current study can be explained by an inflammatory background, and this diagnostic path requires further surveys to be conducted. Despite the novel character of the research, we did not avoid several limitations. The diagnosis of liver steatosis was based on ultrasound examination; liver biopsy and Fibroscan were performed only on the part of patients. Therefore, it was impossible to verify whether the concentration of investigated miRNAs differs in the case of steatosis and steatohepatitis. A future study should be more homogenous according to the enrolled NAFLD patients. Nevertheless, it appears to be the first attempt to examine numerous parameters and relationships between them in the course of NAFLD in a single survey. Thus, the results presented here have a pilot character. Furthermore, in the perspective of continuing this project, it would be indicated that the expression of miRNAs together with direct markers of liver fibrosis should be assessed not only in the serum but also in liver samples to obtain more comprehensive results.

## 5. Conclusions

Verification of the discussed relationships between miRNAs and hematological markers in the course of NAFLD requires further research. The presented observations seem to be valuable due to their innovative nature, as it is difficult to find items in the literature that cover such a large cross-section of various parameters assessed in the context of NAFLD in a single piece of research. A better understanding of the implications between liver steatosis and inflammation could be a strategy for inventing novel markers of NAFLD and its progression.

## Figures and Tables

**Figure 1 biomedicines-13-00230-f001:**
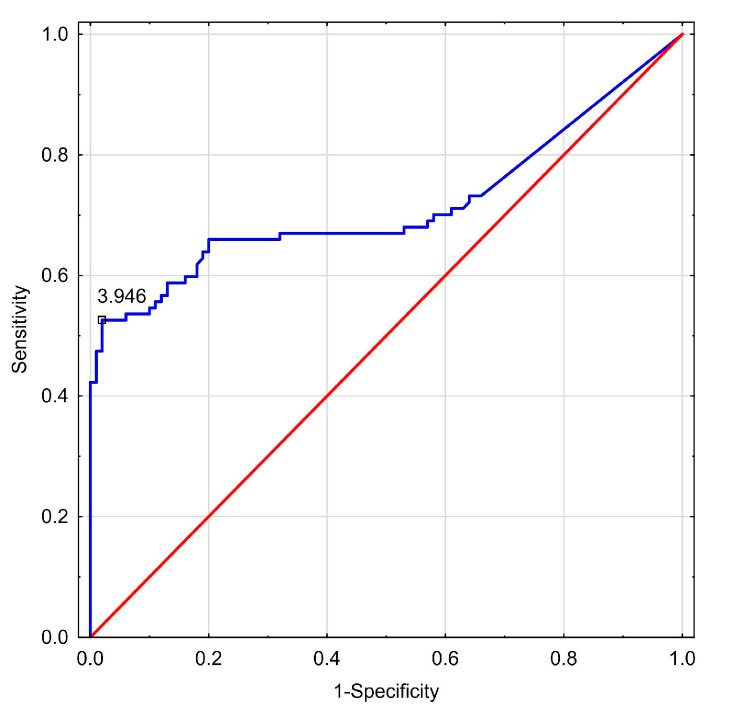
ROC curve (blue line) for miR-126-3p in NAFLD group and random classifier (red line). AUC = 0.716, *p* < 0.0001. A proposed cut-off > 3.95 amol/µL.

**Figure 2 biomedicines-13-00230-f002:**
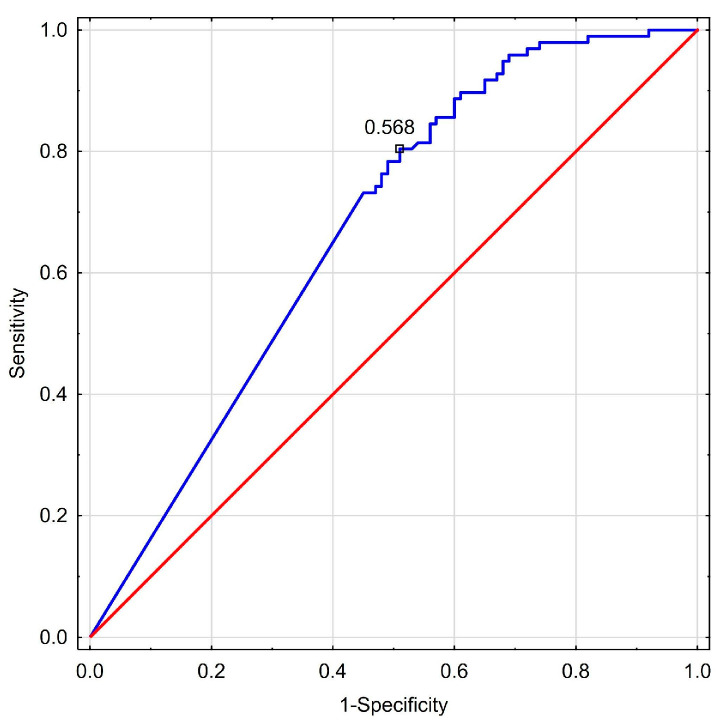
ROC curve (blue line) for miR-197-3p in NAFLD group and random classifier (red line). AUC = 0.691, *p* < 0.0001. A proposed cut-off < 0.57 amol/µL.

**Table 1 biomedicines-13-00230-t001:** Clinical description of study participants.

Parameter	NAFLD n = 97	CONTROLS n = 110	TOGETHER n = 207
sex (m/w)	35/62	57/53	92/115
age (years) (x ± s; me; min—max)	60 ± 15; 62; 22–90	43 ± 15; 39; 20–85	52 ± 15; 54; 20–90
BMI (kg/m^2^) (x ± s; me; min—max)	29.37 ± 4.92; 28.69; 16.26–43.01	22.89 ± 2.38; 23.45; 16.18–36.99	-
type 2 diabetes	27/97	-	-
arterial hypertension	53/97	-	-

**Table 2 biomedicines-13-00230-t002:** Biochemical features of NAFLD patients and the control group.

Parameter [Reference Range]	NAFLD		CONTROLS		*p* [NAFLD vs. CONTROLS]
	me	s	me	s	
albumin [3.2–4.8 g/dL]	3.96	0.56	4.09	0.45	0.004
bilirubin [0.3–1.2 g/dL]	0.9	1.39	0.7	0.27	<0.0001
creatinine [0.5–1.1 g/dL]	0.8	0.29	0.7	0.81	0.001
INR [0.8–1.2]	1.08	0.29	1.08	0.11	0.053
PT [10.4–13 s]	12	3.32	11.7	0.8	0.02
AST [<34 IU/L]	34	60	22	7	<0.0001
ALT [<31 IU/L]	34	128	21	8	<0.0001
GGTP [<31 IU/L]	32	367	21	6	<0.0001
BARD score	2	1	-	-	-
NAFLD fibrosis score	−1.16	1.5	-	-	-

**Table 3 biomedicines-13-00230-t003:** Results of investigated microRNAs in study groups.

Parameter [Reference Range]	NAFLD		CONTROLS		*p* [NAFLD vs. CONTROLS]
	me	s	me	s	
miR-126-3p [amol/μL]	4.38	13.81	0.38	1.22	<0.0001
miR-197-3p [amol/μL]	0.00	1.104	0.39	2.67	0.00003
miR-1-3p [amol/μL]	0.18	1.94	0.00	0.78	<0.0001

**Table 4 biomedicines-13-00230-t004:** Results of determined hematological parameters among study participants.

Parameter [Reference Range]	NAFLD		CONTROLS		*p* [NAFLD vs. CONTROLS]
	me	s	me	s	
PLT [130–400 × 10^9^/L]	233	80	288	61	0.003
MPV [8–11 fL]	7.8	0.88	9	0.88	<0.0001
MPR	0.03	0.03	0.03	0.01	0.81
PDW [40–60%]	55.1	8.17	51.2	5.63	0.0002
PCT [0.12–0.3%]	0.19	0.08	0.21	0.05	0.655
RDW [11–15%]	14.25	2.28	13.38	1.1	0.011
RPR	0.06	0.02	0.05	0.01	0.000003
NEU [2.5–5 × 10^3^/μL]	4.67	2.25	3.71	1.13	0.013
LYM [1.5–3.5 × 10^3^/μL]	1.72	0.78	2.3	0.69	<0.0001
NLR	3.35	2.77	1.78	0.94	<0.0001
PLR	184.27	128.61	138.65	60.42	0.013
RLR	10.74	8.74	6.44	2.29	<0.0001

**Table 5 biomedicines-13-00230-t005:** Results of assessed indirect markers of liver fibrosis in study groups.

Parameter	NAFLD		CONTROLS		*p* [NAFLD vs. CONTROLS]
	me	s	me	s	
AAR	0.96	0.53	1.09	0.38	0.077
APRI	0.46	1.02	0.23	0.12	<0.0001
FIB-4	1.54	1.59	0.72	0.48	<0.0001
GPR	0.44	5.48	0.24	0.09	<0.0001

**Table 6 biomedicines-13-00230-t006:** Results of direct markers of liver fibrosis investigated in the survey.

Parameter	NAFLD		CONTROLS		*p* [NAFLD vs. CONTROLS]
	me	s	me	s	
PICP (ng/mL)	46.08	26.62	44.18	37.39	0.231
PIIINP (ng/mL)	11.00	4.016	10.25	5.61	0.08
PDGF-AB (pg/mL)	26,682.83	7003.17	25,623.2	10,068.8	0.028
TGF-α (pg/mL)	12.09	18.65	24.59	17.21	<0.0001
Laminin (ng/mL)	375.23	231.69	663.27	386.1	<0.0001

**Table 7 biomedicines-13-00230-t007:** Statistically significant correlations of the evaluated miRNAs in NAFLD patients.

NAFLD		
Pair of Markers	R Spearman	*p*
miR-197-3p and PDW	−0.202	0.048
miR-197-3p and PLR	0.212	0.037

**Table 8 biomedicines-13-00230-t008:** AUC values of evaluated indices in NAFLD patients.

Parameter	NAFLD	
	Diagnostic Accuracy of the Marker	
	AUC	*p*
miR-126-3p	0.716	<0.0001
miR-197-3p	0.672	<0.0001
miR-1-3p	-	-
PLT	0.646	0.0002
MPV	0.842	<0.0001
MPR	0.510	0.813
PDW	0.649	0.0001
PCT	0.574	0.065
RDW	0.603	0.009
RPR	0.694	<0.0001
NLR	0.764	<0.0001
PLR	0.600	0.012
RLR	0.740	<0.0001
GPR	0.731	<0.0001
AAR	0.573	0.076
APRI	0.831	<0.0001
FIB-4	0.823	<0.0001
PICP	0.557	0.237
PIIINP	0.583	0.076
PDGF-AB	0.603	0.029
TGF-α	0.774	<0.0001
Laminin	0.767	<0.0001

**Table 9 biomedicines-13-00230-t009:** Specificity, sensitivity, PPV and NPV of the explored parameters among the NAFLD group.

Parameter	NAFLD				
	Cut-Off	Sensitivity [%]	Specificity [%]	PPV [%]	NPV [%]
miR-126-3p [amol/μL]	>3.95	53	98	96	68
miR-197-3p [amol/μL]	<0.57	80	49	60	78
miR-1-3p [amol/μ]	-	-	-	-	-
PLT [130–400 × 10^9^/L]	<230	50	82	71	65
MPV [fl]	<7.9	54	100	100	71
MPR	>0.04	32	79	57	57
PDW [%]	>55.1	51	77	66	64
PCT [%]	<0.17	40	76	59	59
RDW [%]	>12.8	86	34	53	73
RPR	>0.06	63	85	84	65
NLR	>2.02	70	75	72	74
PLR	<237.64	23	93	73	58
RLR	>6.57	76	63	64	75
AAR	>1.04	58	95	92	69
APRI	<0.31	61	59	60	60
FIB-4	>1.18	74	82	80	76
GPR	>0.38	68	84	80	73
PICP (ng/mL)	>67.04	85	32	61	63
PIIINP (ng/mL)	>11.38	51	68	66	52
PDGF-AB (pg/mL)	>19,609.76	91	34	63	74
TGF-α (pg/mL)	<13.37	63	85	84	65
Laminin (ng/mL)	<438.25	67	79	80	66

## Data Availability

Data are contained within the article.

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
