# Peer review of "miRNAs and Hematological Markers in Non-Alcoholic Fatty Liver Disease—A New Diagnostic Path?"

_biomedicines, 2025, doi:10.3390/biomedicines13010230_

Round 1
Reviewer 1 Report
Comments and Suggestions for Authors
Recommendations:
1. We appreciate the efforts but tables should be organized more simpler: Mean(median-if distribution not normal) plus standard deviation or interquartile ranges- plus p-values.
2. Methods section should include information about how statistics was worked up.
3. The micro-RNA levels were standardized to a control (like U6)?: see this: https://doi.org/10.3390/ijms25147898
4. R spearman is a very weal correlation test, moreover, R value is 0,2 so correlation is very weak, couldn't the authors think of a regression model?
5. The AUCs of all markers are modest NOT GOOD, a good value is over 0,8 and the only variable that has over 0,8 is mean platelet volume MPV.
6. You barely mentioned MPV in the discussion section; please assess the importance of thrombocytes in NAFLD see this for help: https://doi.org/10.3390/cimb46080496
7. The iThenticate level still high (30% match).
Author Response
- We appreciate the efforts but tables should be organized more simpler: Mean(median-if distribution not normal) plus standard deviation or interquartile ranges- plus p-values.
Indeed, the included tables contained a lot of data. We simplified them due to your valuable recommendations. In all tables p-values were inserted. Additionally, only medians and standard deviations were left.
- Methods section should include information about how statistics was worked up.
Statistical analysis was described in detail within section 2. Material and methods. Moreover, it was now placed in the subsection 2.1 Statistical analysis.
- The micro-RNA levels were standardized to a control (like U6)?: see this: https://doi.org/10.3390/ijms25147898
For each analyzed miRNA a standard curve was constructed by plotting absorbance values obtained by ELISA miRNA assay against concentrations of miRNAs standards. Concentrations of miRNAs in patients' samples and Quality Control were determined using this standard curve. Protocol was followed according to the manufacturer's instructions.
- R spearman is a very weal correlation test, moreover, R value is 0,2 so correlation is very weak, couldn't the authors think of a regression model?
According to the suggestion of the reviewer linear regression was used to evaluate relationships between continuous variables.
Results (NAFLD patients):
1) miR-197-3p vs PDW
The standardized regression coefficient (b*) = -0.174, p=0.089 (p>0.05)
2) miR-197-3p vs PLR
The standardized regression coefficient (b*) = 0.209, p=0.040
Nevertheless, the achieved results did not show a more powerful character of the obtained relationships, therefore this part of statistical analysis was not included within the manuscript.
- The AUCs of all markers are modest NOT GOOD, a good value is over 0,8 and the only variable that has over 0,8 is mean platelet volume MPV.
Sorry for this overstatement. The description of AUCs related to markers investigated in our study was exaggerated. We improved it within the text.
- You barely mentioned MPV in the discussion section; please assess the importance of thrombocytes in NAFLD see this for help: https://doi.org/10.3390/cimb46080496
Thank you for this comment. We included information related to platelets and MPV to the manuscript according to both: NAFLD and cardiovascular risk. AUC for platelets was provided in the table, as well. Additionally, during the re-editing of the manuscript due to the recommendations of reviewers, it turned out that p value of PDGF-AB is < 0,05, therefore it was also mentioned in the discussion concerning platelets. In the previous version of the manuscript p value of PDGF-AB (NAFLD vs controls) was described as statistically nonsignificant, however, it was our mistake, because according to the statistical analysis it equals 0,028248.
- The iThenticate level still high (30% match).
We looked through the whole manuscript once again and tried to modify certain phrases. The great majority of identified repetitions is present in the section of methodology and is connected with biomarkers and biochemical techniques which were applied by us according to another survey. Secondly, numerous marked overlaps are medical terms/expressions that can not be replaced with other words.
Reviewer 2 Report
Comments and Suggestions for Authors
It is deemed necessary to review Figure 1, which depicts the ROC curve for miR-126-3p, and verify its correspondence with the values reported in Table 9, as the highlighted point does not appear to correspond to the value 3.95. Moreover, it does not seem to align with the x-y intersection at 0.98-0.53 but rather with 0.02-0.53.
Regarding the statement in lines 298–301 ("Probably, investigated herein relationships are related to the inflammatory background of steatosis, too. A common umbrella of inflammation tends to be described as a pathological path shared by vascular disorders, obesity and MASLD"), it is further recommended to also consider other conditions, such as sepsis. A systematic review examining miRNAs involved in liver dysfunction during sepsis has highlighted the upregulation of miR-126 isomers, which are associated with increased apoptosis and reduced cell viability. In this regard, the following article is recommended for further reading: https (https://doi.org/10.3390/ijms23169354)://doi.org/10.3390/ijms23169354 (https://doi.org/10.3390/ijms23169354). I would suggest including a parallel analysis of the miRNAs investigated in your research to understand their variations in other known pathological conditions, thereby making the article more comprehensive and interdisciplinary.
In addition, to ensure a more systematic and coherent presentation, it is advised to revise the Results section by standardizing the order in which results and tables are presented. Specifically, the explanation of results and the corresponding table should consistently appear either before or after the table in question.
Finally, it is recommended to include the institutional affiliation of Author 3.
Author Response
1. It is deemed necessary to review Figure 1, which depicts the ROC curve for miR-126-3p, and verify its correspondence with the values reported in Table 9, as the highlighted point does not appear to correspond to the value 3.95. Moreover, it does not seem to align with the x-y intersection at 0.98-0.53 but rather with 0.02-0.53.
The figures were replaced and now certain values presented on them are visible. On Figure 1 with ROC curve for miR-126-3p the 3.946 value indicates the best cutoff for diagnostic usefulness of this miRNA, at which the sensitivity value reaches 0.526 (indicated on Y axis) and 1-specificity equals 0.02 (indicated on X axis) that gives 0.98 (98%) specificity.
2. Regarding the statement in lines 298–301 ("Probably, investigated herein relationships are related to the inflammatory background of steatosis, too. A common umbrella of inflammation tends to be described as a pathological path shared by vascular disorders, obesity and MASLD"), it is further recommended to also consider other conditions, such as sepsis. A systematic review examining miRNAs involved in liver dysfunction during sepsis has highlighted the upregulation of miR-126 isomers, which are associated with increased apoptosis and reduced cell viability. In this regard, the following article is recommended for further reading: https (https://doi.org/10.3390/ijms23169354)://doi.org/10.3390/ijms23169354 (https://doi.org/10.3390/ijms23169354). I would suggest including a parallel analysis of the miRNAs investigated in your research to understand their variations in other known pathological conditions, thereby making the article more comprehensive and interdisciplinary.
Thank you for this valuable thought. We barely mentioned the inflammation as a possible background of deviations in values of miRNAs in our patients. Thus, it is worth including to the discussion the background of sepsis as another factor related to their expression.
3. In addition, to ensure a more systematic and coherent presentation, it is advised to revise the Results section by standardizing the order in which results and tables are presented. Specifically, the explanation of results and the corresponding table should consistently appear either before or after the table in question.
Thank you for pointing out this item. The presentation of results was standardized.
4. Finally, it is recommended to include the institutional affiliation of Author 3.
The affiliation was included. An editorial mistake occurred in the listing of affiliations. Number 3 was overlooked.
Round 2
Reviewer 1 Report
Comments and Suggestions for Authors
Congratulations!